# Cholesterol-Induced Metabolic Reprogramming in Breast Cancer Cells Is Mediated via the ERRα Pathway

**DOI:** 10.3390/cancers13112605

**Published:** 2021-05-26

**Authors:** Faegheh Ghanbari, Anne-Marie Fortier, Morag Park, Anie Philip

**Affiliations:** 1Division of Plastic Surgery, Department of Surgery, Faculty of Medicine, McGill University, Montreal, QC H3G 1A4, Canada; faegheh.ghanbaridivshali@mail.mcgill.ca; 2Goodman Cancer Research Centre, McGill University, Montreal, QC H3A 1A3, Canada; anne-marie.fortier2@mcgill.ca (A.-M.F.); morag.park@mcgill.ca (M.P.)

**Keywords:** estrogen-related receptor alpha, metabolic reprogramming, breast cancer, cholesterol, obesity

## Abstract

**Simple Summary:**

There is increasing evidence that obesity and high circulating cholesterol levels are associated with an increased risk of recurrence and a higher mortality rate in breast cancer patients via altering the metabolic programming in breast cancer cells. However, the underlying molecular mechanism by which high cholesterol levels reprogram the metabolic pathways in breast cancer cells is not well-understood. We have previously demonstrated that cholesterol acts as an endogenous agonist of estrogen-related receptor α (ERRα), a strong regulator of cellular metabolism. The aim of the current study is to demonstrate whether cholesterol/obesity mediates its pathogenic effect in breast cancer cells via altering metabolic pathways in an ERRα-dependent manner. The findings of this study provide mechanistic insights into the link between cholesterol/obesity and metabolic reprogramming in breast cancer patients and reveal the metabolic vulnerabilities in such breast cancer patients that could be therapeutically targeted.

**Abstract:**

The molecular mechanism underlying the metabolic reprogramming associated with obesity and high blood cholesterol levels is poorly understood. We previously reported that cholesterol is an endogenous ligand of the estrogen-related receptor alpha (ERRα). Using functional assays, metabolomics, and genomics, here we show that exogenous cholesterol alters the metabolic pathways in estrogen receptor-positive (ER+) and triple-negative breast cancer (TNBC) cells, and that this involves increased oxidative phosphorylation (OXPHOS) and TCA cycle intermediate levels. In addition, cholesterol augments aerobic glycolysis in TNBC cells although it remains unaltered in ER+ cells. Interestingly, cholesterol does not alter the metabolite levels of glutaminolysis, one-carbon metabolism, or the pentose phosphate pathway, but increases the NADPH levels and cellular proliferation, in both cell types. Importantly, we show that the above cholesterol-induced modulations of the metabolic pathways in breast cancer cells are mediated via ERRα. Furthermore, analysis of the ERRα metabolic gene signature of basal-like breast tumours of overweight/obese versus lean patients, using the GEO database, shows that obesity may modulate ERRα gene signature in a manner consistent with our in vitro findings with exogenous cholesterol. Given the close link between high cholesterol levels and obesity, our findings provide a mechanistic explanation for the association between cholesterol/obesity and metabolic reprogramming in breast cancer patients.

## 1. Introduction

Obesity and elevated blood cholesterol levels are associated with an increased risk and poor prognosis in breast cancer patients, and this has been linked to profound metabolic alterations that promote tumour growth, progression, and/or response to therapy [1,2]. However, the underlying mechanism is poorly understood. Breast cancer is the second leading cause of death in the United States [3] and is associated with not only dysregulated cell growth but also altered cellular metabolism [4]. Sustained and rapid cell proliferation requires greater accessibility of building blocks to support cell growth and survival under oxidative stress conditions, and breast cancer cells acquire this support through metabolic reprogramming [5,6]. Growing evidence shows that many cancer cells, compared to normal cells, mainly rely on glucose metabolism, using aerobic glycolysis, a process known as the Warburg effect [7]. Even though in this process, ATP generation is less efficient than oxidative phosphorylation (OXPHOS), it invests the carbon skeletons of glucose in several biosynthetic pathways necessary to generate the required molecular building blocks for cell proliferation [7]. Interestingly, it has been shown that intratumoural glucose concentration is minimal, and that oxygen tension is dynamic within tumours [8]. Hence, cancer cells adapt and alter their metabolism according to their environment to align with their high-energy needs, and to produce the building blocks necessary for proliferation and survival. In fact, there is increasing evidence that tumour cells use both glycolysis and mitochondrial oxidative metabolism to satisfy the bioenergetic and/or biosynthetic needs of cancer cells [9,10]. The metabolic alterations that cancer cells adopt are responsive to their environment and as such, glycolysis is often favored in solid breast tumours that experience bouts of hypoxia, and mitochondrial OXPHOS levels can remain active even at low oxygen levels [11,12]. Moreover, under conditions in which cancer cells no longer depend on OXPHOS for ATP production, mitochondrial metabolism can remain an important source of anabolic intermediates [13,14]. On the other hand, it has been demonstrated that breast cancer progression is associated with increased reliance on OXPHOS, as this can favor survival of circulating tumour cells, and promote site-directed metastasis, and resistance to chemotherapy and targeted therapeutics [15,16,17].

The estrogen-related receptor alpha (ERRα) is a transcription factor, which is well-known to regulate mitochondrial OXPHOS, the TCA cycle, and glycolysis [8,18,19,20]. ERRα is expressed in most breast cancer cell types, and its increased activity correlates with unfavorable outcomes in breast cancer patients [8,10,21,22,23,24,25]. It has been reported that knockdown of ERRα in vitro and in vivo significantly inhibits the growth of estrogen receptor-positive (ER+) and triple-negative breast cancer (TNBC) cells [21,26,27,28,29]. Although ERRs have been considered as orphan receptors (i.e., lack endogenous ligands), we and others have recently reported that cholesterol is an endogenous ligand of ERRα and that cholesterol increases ERRα’s transcriptional activity [30,31]. To better understand the mechanism by which obesity and high blood cholesterol levels alter metabolic pathways in breast tumours and promote tumour growth, here, we investigated whether cholesterol, as an agonist of ERRα, promotes alteration in metabolic pathways in breast cancer cells via the ERRα axis. We demonstrate that exogenous cholesterol alters breast cancer cell metabolism in an ERRα-dependent manner. Specifically, cholesterol increases OXPHOS, the TCA cycle, glycolysis, and NADPH levels, while having no impact on the levels of metabolites involved in glutaminolysis, pentose phosphate pathway (PPP), and one-carbon metabolism (OCM) in breast cancer cells. In line with the high cholesterol-induced ERRα-dependent metabolic alterations demonstrated in vitro, we found that obesity (a variable closely linked to high cholesterol) may modulate ERRα metabolic target gene expression in breast cancer patients, detected by analysis of the Gene Expression Omnibus (GEO) database.

## 2. Results

### 2.1. Cholesterol Increases Cellular Proliferation and Aerobic Glycolytic Capacity in Triple-Negative Breast Cancer Cells in an ERRα-Dependent Manner

To demonstrate whether exogenous cholesterol alters cell proliferation in triple-negative breast cancer patient-derived xenograft (TNBC-PDX) cells in an ERRα-dependent manner, these cells were treated with vehicle, cholesterol, and/or Compound 29 (cpd29), a selective inverse agonist of ERRα [32]. Cpd29 has been extensively used to inhibit ERRα transcriptional activity in in vitro and in vivo studies [8,10,33]. As displayed in Figure 1A, cholesterol treatment significantly increases cell proliferation as compared to treatment with vehicle, in TNBC-PDX cells. Importantly, when ERRα was inhibited using cpd29, the enhancing effect of exogenous cholesterol on TNBC-PDX cell proliferation was compromised, with no significant difference between cells treated with both cholesterol and cpd29 when compared to cells treated with cpd29 alone. These results are in agreement with our previous report showing that cholesterol promotes breast cancer cell proliferation in MDA-MB-231 and MCF-7 cells in an ERRα-dependent manner [31].

Next, we examined whether cholesterol regulates aerobic glycolysis in breast cancer cells in an ERRα-dependent manner. For this, we determined the levels of glycolytic metabolites in MDA-MB-231, and glucose uptake and lactate production levels in MDA-MB-231, TNBC-PDX, and MCF-7 cells, in which ERRα was knocked down using siRNA or ERRα activity was blocked with cpd29, followed by treatment with cholesterol. As shown in Figure 1B, cholesterol increases the extracellular acidification rate (ECAR), an indicator of glycolysis capacity, post oligomycin drug injection in MDA-MB-231 cells as determined using the Seahorse XF96. To verify whether this cholesterol-induced effect on ECAR is mediated by ERRα, we knocked down ERRα in MDA-MB-231 cells. As shown in Figure 1B and Appendix A, ERRα was successfully knocked down in MDA-MB-231 cells, and cholesterol-induced ECAR levels were abrogated when ERRα expression was suppressed. This suggests that cholesterol increases the aerobic glycolytic capacity in an ERRα-dependent manner in these cells. Consistent with these findings, determination of glucose consumption and lactate production after cholesterol treatment, using a Profile 400 analyzer, is associated with greater levels of glucose consumption and lactate production, which is linked to glycolysis, in MDA-MB-231 cells. Importantly, cholesterol-induced effects were impaired when ERRα expression was suppressed in these cells (Figure 1C). These results were further validated by measuring the glycolytic intermediate levels in MDA-MB-231 cells, using LC/MS analysis. As displayed in Figure 1D, cholesterol treatment leads to an increasing trend in the indicated glycolytic intermediate levels and significantly enhances the accumulation of glyceraldehyde 3-phosphate (GA3P), fructose-1,6-bisphosphate (F1-.6 bis P), and 2-phosphoglycerate/3-phosphoglycerate (2PG/3PG) metabolite levels. However, when ERRα function was inhibited using cpd29, the cholesterol-induced effect was largely abrogated. We also confirmed that after treating MDA-MB-231 cells with exogenous cholesterol, intracellular cholesterol levels increase approximately 2-fold compared to the vehicle-treated controls (Figure 1E). Moreover, we demonstrated that cholesterol enhances glucose uptake and lactate production levels in TNBC-PDX cells via ERRα (Figure 1F). These data further support the result above, that the cells treated with cholesterol were able to increase their extracellular acidification rate to a higher level than cells treated with vehicle, when challenged with the mitochondrial ATP synthase inhibitor, oligomycin. Together, the data showing that cholesterol increases glucose consumption and lactate production levels in TNBC cells imply that these cells are likely engaging glycolysis at an elevated level in response to exogenous cholesterol. Importantly, our data also demonstrate that this cholesterol-induced effect is mediated via ERRα in MDA-MB-231 and TNBC-PDX cells.

Surprisingly, in MCF-7 cells, cholesterol does not similarly alter glucose uptake or lactate production levels under basal conditions or after ERRα was successfully knocked down in MCF-7 cells (Figure 1G and Appendix A). Based on the TCGA database, the gene expression levels of 3-hydroxy-3-methyl-glutaryl-CoA reductase (HMGCR), a rate-limiting enzyme involved in the cholesterol biosynthesis pathway, are significantly higher in luminal breast tumours compared to triple-negative (TN) breast tumours (Figure 1H). As MCF-7 breast cancer cells are considered as the luminal subtype, it is possible that there are increased levels of intracellular cholesterol in MCF-7 cells and that it may mask the effect of exogenous cholesterol in increasing glucose uptake and lactate production levels in MCF-7 cells. Thus, to decrease intracellular cholesterol levels and sensitize cells to exogenous cholesterol, we used lovastatin (a known HMGCR inhibitor). As shown in Figure 1I, in the presence of lovastatin, cholesterol leads to an increasing trend in glucose consumption and lactate production as compared to the cells treated with vehicle or cholesterol alone in MCF-7 cells. Interestingly, the cells treated with cpd29 demonstrate a significant increase in glucose consumption and lactate production levels, and this suggests that ERRα inhibition using cpd29 increases aerobic glycolysis. Importantly, adding cholesterol does not significantly alter the glucose uptake and lactate production levels. Although adding cholesterol and lovastatin significantly decreases lactate production levels in MCF-7 cells, it does not alter glucose uptake levels in these cells. The observed significant decrease in the presence of cpd29 + cholesterol + lovastatin compared to the cpd29 alone could possibly be due to the impact of lovastatin on other pathways, including inhibition of protein kinase B (AKT)/mammalian target of rapamycin (mTOR) [34,35] in cancer cells. Collectively, these data suggest that cholesterol increases glycolysis rates in triple-negative breast cancer cells, such as MDA-MB-231 and TNBC-PDX cells; however, it does not alter this pathway in ER+ breast cancer cells, such as MCF-7 cells.

### 2.2. Cholesterol Increases OXPHOS Capacity in Breast Cancer Cells via ERRα Pathway

In tumour cells, not only is oxidative phosphorylation (OXPHOS) active even in low oxygen concentrations with the associated ATP generation facilitating cellular growth, but upregulated OXPHOS causes chemotherapeutic resistance [11,12,36]. Thus, it was interesting to determine whether cholesterol regulates the OXPHOS capacity in breast cancer cells via ERRα. As shown in Figure 2A–C, cholesterol significantly enhances the expression of ERRα-induced OXPHOS target genes (NDUFB7, ATP5L, and COX5B) in MDA-MB-231, MCF-7, and TNBC-PDX cells. However, when ERRα action was inhibited using cpd29, the cholesterol-induced effect was compromised in these breast cancer cells, demonstrating that ERRα mediates the cholesterol effect on those OXPHOS target genes, except for the expression of COX5B in MCF-7 cells, where it was significantly higher in the presence of cholesterol while ERRα was inhibited. These data suggest that cholesterol may increase the expression of COX5B via pathways other than ERRα in MCF-7 cells. In addition, we demonstrated that cholesterol enhances the expression of ERRα (encoded by the *ESRRA*), and that in the presence of cpd29, ERRα mRNA levels significantly decrease, and that cholesterol does not rescue cpd29′s inhibitory effect on MDA-MB-231, MCF-7, and TNBC-PDX cells (Appendix A). It is important to mention that based on TCGA datasets, there is a positive correlation between the expression of ERRα and that of NDUFB7 (R^2^ = 0.32), COX5B (R^2^ = 0.34), and mitochondrial ATP synthetase subunit delta (ATP5D) (R^2^ = 0.31) (the enzymes involved in the electron transport chain (ETC)). This positive correlation is particularly significant in basal-like breast tumours (studied in vitro using MDA-MB-231 and TNBC-PDX cells) and luminal breast tumours (studied in vitro using MCF-7 cells) (Appendix A). This supports the findings from other studies that ERRα modulates the enzymes involved in OXPHOS in breast tumours [8,18].

Since we observed that exogenous cholesterol increases ERRα-induced target genes involved in OXPHOS pathway, it was interesting to verify if these alteration in genes expression levels have any significance on oxygen consumption rate, which is associated with the OXPHOS pathway. As shown in Figure 2D, cholesterol significantly increases the maximal respiration, which is associated with a greater spare capacity compared to the vehicle-treated one in MDA-MB-231 cells, suggesting that cholesterol increases oxidative capacity and the ability of MDA-MB-231 cells to respond to increased energy demand or stress conditions. To verify whether this cholesterol-induced effect on OXPHOS rate is mediated by ERRα, we knocked down ERRα in MDA-MB-231 cells. As demonstrated in Figure 2D, the cholesterol-induced increase in maximal respiration and spare capacity was compromised when ERRα was knocked down, suggesting that this cholesterol stimulatory effect is ERRα-dependent. Together, these findings suggested that exogenous cholesterol increases OXPHOS capacity in breast cancer cells, and that this effect is mediated via ERRα axis. Furthermore, the results showing that cholesterol enhances both OXPHOS and aerobic glycolysis indicate that cholesterol enables metabolic flexibility in breast cancer cells.

### 2.3. Cholesterol Enhances the Abundence of Certain Key Enzymes and Metabolites of the TCA Cycle in an ERRα-Dependent Manner

As an upregulated TCA cycle is involved in biomass generation and helps cancer cell growth [9,10,36,37], it was interesting to determine whether cholesterol regulates TCA cycle intermediate levels in an ERRα-dependent manner in breast cancer cells. As shown in Figure 3A–C, cholesterol induces the expression of ACO2, CS, and FH, the ERRα metabolic target genes involved in the TCA cycle, in MDA-MB-231, MCF-7, and TNBC-PDX cells. Importantly, blocking ERRα activity using cpd29 compromises this cholesterol-induced effect. The functional significance of these alterations in gene expression levels was verified by determining the levels of TCA cycle metabolites using steady-state metabolomics. As shown in Figure 3D, cholesterol significantly augments aconitic acid and fumaric acid levels in MDA-MB-231 cells, although the increasing trend observed in the levels of the other TCA cycle intermediates was not statistically significant. Furthermore, when ERRα activity was inhibited using cpd29, no significant differences were observed in the TCA cycle intermediate levels in the presence of cholesterol. Overall, these results suggest that cholesterol primes breast cancer cells for increased levels of mitochondrial oxidative metabolism, and that this is ERRα-dependent. Furthermore, these findings are in agreement with the above results that cholesterol increases OXPHOS capacity in breast cancer cells via the ERRα pathway, as the TCA cycle is intrinsically linked to OXPHOS.

### 2.4. Cholesterol Does Not Alter the Abundances of Metabolites in the Glutaminolysis Pathway in Breast Cancer Cells

It has been shown that glutaminolysis is upregulated in most of the transformed cells, and is critical for cancer cells to survive under oxidative stress conditions [8,38,39]. Therefore, it was interesting to assess the impact of cholesterol on the glutamine metabolism in MDA-MB-231, MCF-7, and TNBC-PDX cells. As shown in Figure 4A, no alterations were observed in the levels of glutamine uptake and glutamate excretion in the presence of cholesterol. Interestingly, when ERRα expression was suppressed, there was a significant increase in the above-mentioned metabolite levels; however, the presence of cholesterol does not alter their levels. These data were further confirmed by measuring the intracellular glutamine and glutamic acid levels in MDA-MB-231 cells (Figure 4B). As shown in Figure 4B, cholesterol does not alter intracellular glutamine and glutamic acid levels in these cells. Consistently, these findings that cholesterol does not alter glutamine uptake and glutamate excretion levels were observed in MCF-7 and TNBC-PDX cells (Figure 4C,D). Of note, converting the data to the ratio of intracellular glutamate to glutamine does not show any changes among the treatment groups. Our results show that in the absence of ERRα activity, we observed a significant increase in glutamine uptake and glutamate excretion levels, suggesting that ERRα is an important regulator of glutamine metabolism. Together, our data suggest that although lack of ERRα increases the abundances of metabolites involved in the glutaminolysis pathway, exogenous cholesterol does not alter the glutaminolysis metabolite levels in breast cancer cells.

### 2.5. Cholesterol Does Not Alter the Abundances of Metabolites in the Pentose Phosphate Pathway (PPP) and the One-Carbon Metabolism (OCM) Pathway in TNBC Cells

It has been shown that the PPP and OCM pathways are involved in nucleotide and NADPH generation in cancer cells for survival under oxidative stress conditions [8,38,39]. Thus, we next analyzed the effect of cholesterol on these two pathways. We first determined the relative gene expression levels of G6PD and 6PGD, two critical enzymes involved in NADPH production in the pentose phosphate pathway (Figure 5A). As shown in Figure 5A, cholesterol significantly decreases the expression of G6PD, while it significantly increases the 6PGD expression in TNBC-PDX cells. This raises the possibility that cholesterol increases the expression of 6PGD to maintain NADPH homeostasis in breast cancer cells. Interestingly, inhibiting ERRα activity using cpd29 significantly increases the expression of G6PD and 6PGD, and the cholesterol-induced alteration of these two enzymes is abrogated in the presence of cpd29. These data suggest that the effect of cholesterol on the expression of these enzymes is ERRα-dependent. We further verified the expression of MTR and GART, two key enzymes involved in the OCM pathway. As demonstrated in Figure 5B, exogenous cholesterol significantly decreases the expression of GART, while it does not significantly alter the expression of MTR. Moreover, in the presence of cpd29, cholesterol does not significantly alter the expression levels of MTR and GART in TNBC-PDX cells.

To verify whether the cholesterol effect on the above-mentioned gene expression levels leads to alterations in the PPP and OCM pathway metabolite levels, we performed steady-state metabolomics. As shown in Figure 5C, although exogenous cholesterol induces a small increase in PPP metabolite levels, its stimulatory effect is not significant, as seen in MDA-MB-231 cells. In ERRα-inhibited breast cancer cells, a significant increase in ribulose-5p was observed, and it was seen that adding exogenous cholesterol does not significantly alter the levels of metabolites. We also analyzed the OCM pathway metabolites and, as shown in Figure 5D, cholesterol does not significantly alter OCM metabolite levels. However, inhibition of ERRα activity using cpd29 significantly increases the levels of several OCM intermediates, including phospho-serine (p-serine), taurine, and histidine, in MDA-MB-231 cells. However, exogenous cholesterol does not significantly alter the OCM metabolite levels, regardless of the presence or absence of cpd29. Although in the presence of cpd29, cholesterol demonstrates a decreasing pattern in p-serine levels compared to the cpd29 alone, it is not statistically significant. Overall, our findings align with other studies which suggested that ERRα acts as a suppressor of the PPP and the OCM pathways [8,38]. Consistent with this finding, cholesterol, as an agonist of ERRα, significantly decreases the expression of G6PD and GART in an ERRα-dependent manner and does not significantly alter the abundances of metabolites involved in the PPP and the OCM pathways, with the exception that cholesterol significantly increases 6PGD expression levels, possibly to maintain NADPH homeostasis.

### 2.6. Cholesterol Increases NADPH Levels in Breast Cancer Cells

Our findings so far have demonstrated that exogenous cholesterol promotes cellular proliferation and increases mitochondrial metabolism, glycolysis capacity, and the expression of 6PGD involved in NADPH production. Thus, it was of interest to assess the impact of cholesterol on intracellular NADPH levels in breast cancer cells, as NADPH is involved in several reductive biosynthesis reactions and reactive oxygen specious (ROS) mitigation [33,40,41]. Our results demonstrate that cholesterol significantly augments NADPH levels compared to the vehicle-treated controls in MDA-MB-231 and MCF-7 (Figure 6A,B). However, when ERRα expression was blocked using siRNA, the cholesterol-induced increase in NADPH levels was abrogated. Together, these results suggest that cholesterol increases NADPH levels in breast cancer cells and that this cholesterol-induced effect is mediated via ERRα.

### 2.7. Expression of the Metabolic Target Genes of ERRα Is Increased in the Basal-Like Primary Breast Tumours Obtained from Overweight/Obese Breast Cancer Patients Compared to Those Obtained from Lean Patients

As obesity is a variable closely linked to high cholesterol levels, we next analyzed the ERRα metabolic gene signature profile in basal-like primary breast tumours (considered to be representative of the TNBC subtype) from overweight/obese patients versus those from lean patients, using data available at the GEO database. As shown in Figure 7A, we observed that the expression of ERRα is higher in the basal-like breast tumours of overweight/obese patients compared to those of lean patients. Interestingly, the expression of several ERRα metabolic target genes involved in OXPHOS are significantly elevated in the basal-like breast tumours of overweight/obese patients compared to those of the lean patients (Figure 7B). In addition, we observed a significant increase in the expression of FH, and an increasing trend in the expression of isocitrate dehydrogenase 2 (IDH2), CS, and ACO2, all of which are involved in the TCA cycle (Figure 7C). Our results also show that even though there is an increasing trend in the expression of several ERRα target genes involved in the glycolytic pathway, such as lactate dehydrogenase B (LDHB) and hexokinase-2 (HK2), no significant alteration was seen in their gene expression levels in the basal-like breast tumours of overweight/obese patients compared to the lean patients (Figure 7D). In addition, we demonstrated that in the basal-like breast tumours of overweight/obese breast cancer patients, the expression levels of ERRα target genes involved in glutaminolysis remained unchanged compared to those in the lean patients (Figure 7E). However, the expression of G6PD and GART, enzymes associated with the PPP and the OCM pathway (respectively), shows a significant increase in overweight/obese breast cancer patients compared to the lean patients (Figure 7F,G). Furthermore, we observed a significant increase in the expression of GSTM1 and SOD2, two key enzymes related to ROS detoxification, in the basal-like breast tumours of overweight/obese patients compared to those of lean patients (Figure 7H). Together, these data suggest that the ERRα metabolic target genes, such as genes involved in OXPHOS, the TCA cycle, and ROS detoxification, and key enzymes involved in PPP and purine biosynthesis pathways, such as G6PD and GART, respectively, are higher in the basal-like breast tumours of overweight/obese breast cancer patients compared to those of lean patients. These data are consistent with, and are validated by, the in vitro data using breast cancer cell lines, such as MDA-MB-231, MCF-7, and/or TNBC-PDX, presented in the current study and in a previous publication by our group [31].

## 3. Discussion

Obesity and high circulating cholesterol levels have been associated with an increased likelihood of recurrence and a higher mortality rate in breast cancer patients [42,43,44,45,46,47] and have been linked to marked alterations in the metabolic pathways in breast cancer cells [1,2]. However, the underlying mechanism by which elevated cholesterol levels alter the metabolic pathways in breast cancer cells is not well understood. While there are several studies demonstrating that high cholesterol intake does not increase the plasma cholesterol levels [48], suggesting that exogenous cholesterol (dietary cholesterol) decreases de novo cholesterol synthesis, in order to maintain cholesterol balance, there are also cohort studies reporting that there is a linear relationship between dietary cholesterol and total plasma cholesterol levels [49,50,51]. A recent report demonstrated that the link between dietary cholesterol and breast cancer recurrence was significant when cholesterol consumption was higher than 370 mg/day [45]. In accordance with this notion, several epidemiological studies showed an association between dietary cholesterol consumption and the increase in the risk of both ER+ and TN breast cancer [45,46,47,52,53,54,55]. Additionally, it has been reported that mice bearing breast cancer xenografts demonstrate significantly greater tumour proliferation and angiogenesis when fed with a HF/HC diet [56,57,58], and that in such mice, ezetimibe, an inhibitor of intestinal cholesterol uptake, abolishes the growth of breast cancer xenografts [58]. This finding supports the notion that cholesterol itself can impact tumour pathophysiology. In addition, altered metabolism of cholesterol has been shown to be involved in resistance to tamoxifen therapy in ER+ breast cancer [56,59]. Taken together, there is ample evidence to support the conclusion that high blood cholesterol level and altered cholesterol metabolism are risk factors in the onset, recurrence, or resistance to therapy in breast cancer. Given that we and another group have shown that cholesterol acts as an endogenous ligand of ERRα and increases its transcriptional activity and cellular growth [30,31], it was of interest to elucidate the metabolic pathways by which the cholesterol-ERRα axis may mediate its pathogenic effect in breast cancer cells.

In alignment with our previous report that cholesterol promotes cellular growth in MDA-MB-231, triple-negative, and MCF-7, ER+ breast cancer cells [31], here, we show that cholesterol enhances the proliferation of TNBC-PDX cells, and demonstrate that the growth stimulatory effect of cholesterol is ERRα-dependent in TNBC-PDX cells as in the other two breast cancer cell lines studied. The mechanism by which cholesterol promotes cellular growth in breast cancer cells may involve cholesterol acting as an agonist of ERRα, and enhancing the interaction of ERRα with its coactivator PGC-1α, as we have previously demonstrated [31]. This enhanced interaction, in turn, results in the expression of ERRα target genes including its own gene expression (auto-induction) [31,60], leading to the induction of a cascade of metabolic pathways, such as aerobic glycolysis, and oxidative metabolism (OXPHOS and the TCA cycle).

An important finding in the current study is that exogenous cholesterol enhances both mitochondrial oxidative metabolism (the expression of key enzymes and the abundances of metabolites involved in the TCA cycle, and OXPHOS) and aerobic glycolysis in TNBC cells. This is distinct from the finding in ER+ breast cancer cells, where cholesterol increased mitochondrial oxidative metabolism in a similar manner as in TNBC while aerobic glycolysis remained unaltered. Interestingly, we found that obesity (a variable linked to high cholesterol levels [61]) may modulate the ERRα metabolic gene expression profile in breast cancer patients in a manner consistent with the high cholesterol-induced ERRα-dependent metabolic alterations in vitro. For the analysis of the ERRα metabolic gene expression profile in breast cancer patients, we used the data obtained from the Gene Expression Omnibus (GEO) database to compare the basal-like breast tumours of overweight/obese versus lean patients, based on the premise that basal-like breast tumours are considered to be representative of the TNBC, a subtype for which there is no satisfactory treatment available. Additionally, ERRα has been shown to be overexpressed in the TNBC subtype [26,62]. Our results show that the expression of ERRα metabolic target genes involved in OXPHOS, the TCA cycle, and enzymes important for detoxification are increased in obesity (basal-like breast tumours of overweight/obese versus lean patients), which aligns with our in vitro findings that exogenous cholesterol enhances the above-mentioned enzyme levels in TNBC cells. Together, these findings provide key insights into the mechanism underlying the association between cholesterol/obesity and metabolic reprogramming in breast cancer patients. In addition, our results reveal metabolic vulnerabilities in high cholesterol/obese/overweight patients, which are molecular pathways that may be therapeutically targetable. Accumulating evidence demonstrates that most oncogenes enhance aerobic glycolysis and that this increased reliance on glycolytic metabolism is an inherent property of the transformed cells [63]. Interestingly, our finding that cholesterol enhances the levels of glycolytic metabolites in TNBC cells, but not in ER+ breast cancer cells, aligns with other studies reporting that enhanced aerobic glycolysis is positively correlated with the malignancy of tumour cells [63,64]. Remarkably, our observation (based on the data available at the TCGA database) that the mRNA level of HMGCR, a key enzyme involved in the cholesterol biosynthesis pathway [65], is overexpressed in the luminal subtype of breast tumours, is in line with our finding in ER+ breast cancer cells, that treatment with lovastatin (a known inhibitor of HMGCR) leads to a trend towards decreased lactate production levels, which is associated with glycolysis. This decreasing trend in lactate production levels using lovastatin could possibly be due to a decrease in intracellular cholesterol levels in these cells [66]. This finding raises the possibility that decreasing intracellular cholesterol levels may decrease glycolysis also in ER+ breast cancer cells, as in TNBC cells.

There is growing evidence to demonstrate that mitochondria produce up to 90% of the generated ATP in some cancer cells and that OXPHOS is active even at 0.5% oxygen levels [11,12,36]. Several studies also reported that resistance to the K-RAS inhibitor in pancreatic cancer [17], and BRAF inhibitors in melanoma [15], is associated with such a shift to oxidative metabolism. Importantly, our findings demonstrate that exogenous cholesterol increases not only aerobic glycolysis rates, but also mitochondrial oxidative metabolism (TCA cycle intermediate accumulation and the OXPHOS rates) via the ERRα axis in breast cancer cells. Our data, that suggest cholesterol induces the expression of key enzymes involved in OXPHOS and the TCA cycle via ERRα, are in line with other studies reporting that ERRα is the master regulator of mitochondrial metabolism and is associated with the upregulation of enzymes involved in OXPHOS and the TCA cycle [67,68]. These studies are in accord with our finding that cholesterol, as an ERRα agonist, enhances mitochondrial respiration and the accumulation of TCA cycle intermediates. The upregulation in both glycolysis and oxidative metabolism levels has been shown to be associated with increased NADPH levels and metabolic flexibility, which cause breast cancer cells to proliferate faster and help them to survive oxidative stress conditions [37]. It has also been reported that enhanced OXPHOS is linked to resistance to chemotherapeutics in certain cancers [11,12,69]. Notably, our in vitro data, suggesting cholesterol enhances oxidative metabolism in vitro are in agreement with the in vivo data obtained in the current study using obesity as a variable, i.e., by analyzing basal-like breast tumours from overweight/obese versus lean breast cancer patients.

It has been shown that the glutaminolysis, PPP, and OCM pathways are critical for cancer cells to generate nucleotides, nucleic acids, and NADPH, which are required for cancer cells to survive under stressed conditions [8,38,39]. Interestingly, our data demonstrate that exogenous cholesterol, as an agonist of ERRα, does not significantly alter the abundances of metabolites corresponding to the glutaminolysis, PPP, and OCM pathways in TNBC or ER+ breast cancer cells. Our results, demonstrating that decreasing ERRα activity increases glutamine uptake and glutamate excretion in breast cancer cells, are in agreement with the previous reports that ERRα acts as a suppressor in glutamine oxidation, as well as the PPP and OCM pathways [8,38]. Furthermore, a recent study revealed that the increase in glutamine metabolism upon ERRα inhibition using cpd29 treatment, as we have found in the current study, is linked to reduced glutathione (GSH) production, which is involved in ROS elimination, and thus in the survival of breast cancer cells under oxidative stress [33].

Our finding, that cholesterol decreases the expression of G6PD, a key enzyme involved in NADPH synthesis in the PPP pathway [41], and the expression of GART, an enzyme involved in the de novo purine synthesis (OCM-related) pathway [38] in an ERRα-dependent manner, is in agreement with other studies demonstrating that ERRα downregulates the expression of enzymes involved in the OCM pathway [38]. The cholesterol-induced and ERRα-dependent increase in the expression of 6PGD, a third enzyme of the PPP involved in NADPH synthesis [40], although a surprising finding, could be an adaptive response to maintain NADPH homeostasis in breast cancer cells, as NADPH is utilized for rapid tumour growth and survival under excessive oxidative stress conditions [40]. Interestingly, our in vitro finding with exogenous cholesterol contradicts the data for the expression of G6PD and GART in the basal-like breast tumours of overweight/obese breast cancer patients. In TNBC-PDX cells, exogenous cholesterol downregulates the expression of the above-mentioned enzymes in an ERRα-dependent manner. However, obesity (a variable linked to high cholesterol levels) is associated with a higher expression of G6PD and GART enzymes, as detected in the basal-like breast tumours of overweight/obese versus lean breast cancer patients. The discrepant expression levels of G6PD and GART between TNBC-PDX cells treated with cholesterol and overweight/obese basal-like breast cancer patients are possibly due to multiple factors, as obesity is a complex disease, with upregulation of several growth factors and activation of other signaling pathways, such as the PI3K/Akt/mTOR pathways [70,71].

In cancer cells, overcoming oxidative stress is a critical step for tumour progression. NADPH homeostasis plays a critical role in the ROS detoxification system by reducing oxidized glutathione (GSSG) to reduced glutathione (GSH), which is essential for mitigating ROS production during cell proliferation [33,40,41]. In addition, NADPH is a crucial electron source for reductive biomass synthesis, such as nucleotides, fatty acids, and amino acids, to sustain rapid tumour growth [72,73]. Our intriguing finding, demonstrating that exogenous cholesterol increases NADPH levels in breast cancer cells in an ERRα-dependent manner, is physiologically relevant, as ERRα is known to be involved in NADPH generation and ROS elimination via the induction of key enzymes involved in ROS detoxification, such as SOD2 and GSTM1, which have been identified as target genes of ERRα [20]. This is in agreement with the data in the current study linking obesity (associated with high blood cholesterol levels) to a higher expression of SOD2 and GSTM1, as detected by an increase in the levels of those two enzymes and ERRα in overweight/obese basal-like breast cancer patients compared to lean patients. Additionally, these results are in line with our previous finding that cholesterol induces the expression of the above two key detoxification enzymes, SOD2 and GSTM1, in breast cancer cells, and that this effect is mediated via ERRα [31].

In summary, the findings from the current study reveal that the mechanism by which cholesterol promotes cellular growth in breast cancer cells involves cholesterol acting as an agonist of ERRα. We have previously reported that cholesterol enhances the interaction of ERRα with its coactivator PGC-1α [31]. Here, we extend those findings to show that this enhanced interaction leads to increased expression of ERRα metabolic target genes, including ESSRA itself (specific auto-induction) [31,60]. This process induces a cascade of metabolic pathways, such as aerobic glycolysis, oxidative metabolism (OXPHOS and the TCA cycle), and the expression of 6PGD involved in the pentose phosphate pathway. These upregulated pathways have been shown to be involved in increased anabolic intermediates and in electron acceptors that are used in the electron transport chain (ETC) to provide the transformed cells with their bioenergetic and/or biosynthetic needs [9,10,36,37]. The above-mentioned pathways are also involved in increasing NADPH levels possibly via malate-aspartate shuttle [8] and via upregulation of the 6PGD enzyme involved in the PPP [40] in the presence of exogenous cholesterol. These enhanced NADPH levels could potentially increase the biomass synthesis and ROS detoxification in breast cancer cells to promote cellular proliferation [74]. In addition, it is interesting to speculate that the upregulated oxidative metabolism and detoxification enzyme expression observed in obesity (a variable linked to high blood cholesterol levels), as detected in the basal-like breast tumours of overweight/obese versus lean breast cancer patients, is possibly mediated via the cholesterol–ERRα axis, and that this upregulation increases cell proliferation and may cause resistance to chemotherapeutics and some targeted therapies [15,16,17,69]. Together our findings above provide key mechanistic insights into the link between cholesterol/obesity and metabolic reprogramming leading to increased cellular proliferation in breast cancer patients, and reveal the metabolic vulnerabilities in such breast cancer patients that could be therapeutically targeted.

## 4. Materials and Methods

### 4.1. Cell Culture

The MDA-MB-231 and MCF-7 cell lines were obtained from Dr. Sylvie Mader (Université de Montréal, Montréal, QC, Canada). The above-mentioned cell lines were cultured in DMEM supplemented with 10% fetal bovine serum (FBS) and 1% penicillin/streptomycin. The TNBC-PDX cell line GCRC1887 was obtained from the breast tissue and data bank at the Goodman Cancer Research Centre—Research Institute of McGill University Health Centre (MUHC) (Montréal, QC, Canada) supported by the Réseau de Recherche en Cancer of the Fonds de Recherche du Québec-Santé and the Quebec Breast Cancer Foundation (Montréal, QC, Canada). Banking of human specimens and associated clinical data were approved by MUHC research ethics board (study approval SUR-2000-966 and SUR-99-780). All patient data and biological samples were obtained from patients at the MUHC after obtaining informed consent. These cells were maintained in DMEM/F12 (3:1), 5% FBS, hydrocortisone 0.4 µg/mL (Sigma (Mississauga, ON, Canada), H0888-5G), recombinant human epidermal growth factor (EGF) 10 ng/mL (AF-100-15, PeproTech (Rocky Hill, NJ, USA)), insulin 5 µg/mL (12585-014, Gibco (Mississauga, ON, Canada)), Y-27632 dihydrochloride, Rho inhibitor 10 µM (ab120129, Abcam (Cambridge, MA, USA)), prostaglandin E2 (PGE2) 1 µM (S3003, Selleckchem (Burlington, ON, Canada)), gentamicin 50 µg/mL (15710-072, Gibco (Mississauga, ON, Canada)), pen/strep 1× (Sigma (Mississauga, ON, Canada), 15140-122), and fungizone 0.5× (15290-026, Invitrogen (Burlington, ON., Canada)). Experimental design for cholesterol and/or cpd29 treatment was as follows: MDA-MB-231, MCF-7, and TNBC-PDX cells were cultured in their respective culture media, and then the cells were switched 24 h before treatments to their respective basic media with no phenol red, supplemented with 2% FBS that was lipoprotein depleted, and charcoal-stripped [75]. Lipoprotein-depleted FBS was purchased from Kalen Biomedical LLC (Germantown, MD, USA) and charcoal-stripped to remove steroid hormones. Then, these cells were treated with cholesterol and/or cpd29 for 24 or 48 h before analysis.

Cholesterol-water soluble (C4951-30MG) was purchased from Millipore Sigma. Lovastatin (sc-200850A, Santa Cruz Biotechnology (Santa Cruz, CA, USA)), a known cholesterol-lowering drug, was used to decrease intracellular cholesterol levels. Compound 29 (cpd29), a known synthetic inverse agonist of ERRα, was used to decrease ERRα transcriptional activity, and it was a generous gift from Dr. Donald McDonnell (Duke University, Durham, NC, USA).

### 4.2. Real-Time Metabolic Analysis

Multiparameter metabolic analysis of MDA-MB-231 cells was performed simultaneously in the Seahorse XF96 extracellular flux analyzer (Seahorse Bioscience, Agilent (Santa Clara, CA, USA)). Briefly, MDA-MB-231 cells were transfected with either siRNA-control (si-CTL) or siRNA-ERRα (si-ERRα) for 48 h, followed by treatment with vehicle or 5 µM cholesterol for 24 h. On the day of the assay, the treated MDA-MB-231 cells were plated on XF96 (20,000 cells per well), and the culture medium was replaced with Seahorse base media DMEM (supplemented with 2 mM glutamine, 2 mM pyruvate, and 12.5 mM Glucose, pH 7.4) 1 h before the assay and for the duration of the experiment. Mitochondrial complex inhibitors were prepared based on the mitochondria stress kit (Agilent (Santa Clara, CA, USA), 103015-100) instructions. After establishing the baseline oxygen consumption rate (OCR) and extracellular acidification rate (ECAR) readings, mitochondria inhibitors (oligomycin, FCCP, and rotenone/antimycin) were injected accordingly, and OCR and ECAR were measured.

### 4.3. Metabolites Measurement

Cellular glucose, glutamine, lactate, glutamate, and ammonium were measured using the Bio-Profile 400 analyzer (Nova Biomedical Corp., Waltham, MA, USA). MDA-MB-231 and MCF-7 cells were transfected with either siRNA-control (si-CTL) or siRNA-ERRα (si-ERRα) for 48 h, followed by treatment with vehicle or 5 or 10 µM cholesterol for 24 h, respectively. In addition, for MCF-7 cells, the cells were treated with 10 µM cholesterol and/or lovastatin and/or cpd29 for the duration of 48 h. TNBC-PDX cells were treated with vehicle, cholesterol (10 µM), and/or cpd29 (10 µM) for 48 h. The media were then removed and centrifuged at 15,000 rpm for 10 min to remove the cell debris, and the media were maintained on ice until analysis. Glucose or glutamine uptake was calculated as the differences in glucose or glutamine content between culture media and unseeded media incubated in parallel plates. Lactate, glutamate, and ammonium production were reported as measured using the instrument, and all the data were normalized for cell count and respective vehicle.

### 4.4. Metabolomics

Metabolic profiling was performed in a metabolomics core facility located at the Rosalind and Morris Goodman Cancer Research Centre at McGill University (Montréal, QC, Canada), using gas chromatography–mass spectrometry (GC/MS) and liquid chromatography–mass spectrometry (LC/MS). MDA-MB-231 cells were plated in 10 cm dishes and were treated with vehicle, cholesterol, and/or cpd29 with a concentration of 5 µM for 48 h. For GC/MS, the cells were rinsed in saline, quenched in 80% HPLC-grade methanol, sonicated, centrifuged, and the supernatants were dried in a cold trap (Labconco (Kansas City, MO., USA)) overnight at −1 °C. Pellets were solubilized in methoxy-amine HCl, incubated at room temperature for one hour, and derivatized with MTBSTFA at 70 °C for one hour. Next, 1 µL was injected into an Agilent 5975C GC/MS in SCAN mode, and the data were analyzed using Masshunter software (Agilent Technologies) [76]. LC/MS sample preparation was performed according to the core facility protocol [25]. Briefly, cells were rinsed in 150 mM ammonium formate (Sigma (Mississauga, ON, Canada)) and extracted using 230 µL of LC/MS-grade 50% methanol/50% water mixture and 220 µL of cold acetonitrile. Samples were then homogenized and centrifuged. The upper aqueous layer was dried by vacuum centrifugation (Labconco (Kansas City, MO., USA)). Samples were separated by UHPLC (ultra high-performance liquid chromatography) (1290 Infinity, Agilent Technologies, (Santa Clara, CA, USA)). Then, metabolites were eluted into an electrospray ionization source (ESI) and detected by multiple reaction monitoring (MRM) using a triple quadrupole mass spectrometer (6430 QQQ, Agilent Technologies, (Santa Clara, CA, USA)).

### 4.5. siRNA Transfection

As it was previously described [77], siRNAs directed against ERRα (Invitrogen, AM16708/289481) with the sense sequence 5′-CCGCUUUUGGUUUUAACC-3′ and antisense sequence 5′-GGUUUAAAACCAAAAGCGG-3′ or control scrambled siRNAs (Invitrogen, AM4611, negative control) were transfected into MCF-7 and MDA-MB-231 cells using Lipofectamine RNAiMAX Transfection Reagent (Invitrogen, Burlington, ON, Canada) following the manufacturer’s instructions. After 48 h post-transfection, fresh phenol red-free medium containing 2% lipoprotein-depleted and charcoal-stripped serum was added, and cells were treated with cholesterol (5 µM for MDA-MB-231, and 10 µM for MCF-7 cells). The knocked-down ERRα breast cancer cells were used for metabolic assays.

### 4.6. Immunoblotting

To determine whether ERRα was successfully knocked down in MCF-7 and MDA-MB-231 cells, the cell lysates were subjected to immunoblotting. ERRα levels were detected using the rabbit monoclonal anti-ERRα antibody (ab76228), and the mouse monoclonal anti-alpha tubulin antibody (ab7291) was used to detect alpha-tubulin as a loading control. These antibodies were purchased from Abcam (Cambridge, MA, USA). Image-J software (version 1.51, National Institutes of Health (NIH), Bethesda, MD, USA) was used for densitometric analysis of immunoblots.

### 4.7. RNA Preparation and Analysis

Total RNA was extracted using an RNeasy mini kit (74104, Qiagen (Germantown, MD, USA)). One microgram of total RNA was used for the first-strand synthesis with a high-capacity cDNA reverse transcription kit (4368814, Life Technologies (Grand Island, NY, US)). Real-time PCR was performed using the BrightGreen qPCR master mix (ABMMastermix-R, Diamond (Richmond, BC, Canada)) with gene-specific primers. The sequences of the primers included in this study are included in Appendix A. Real-time PCR was performed on the 7500 real-time PCR system (Applied Biosystems (Richmond, BC, Canada)). Results were quantified using the 2^−ΔΔ*CT*^ method and were normalized to the endogenous control, glyceraldehyde 3-phosphate dehydrogenase (GAPDH).

### 4.8. NADPH Quantification Assay

The intracellular NADPH levels were measured using the NADP^+^/NADPH assay kit (Abcam (Cambridge, MA, USA), ab65349). MDA-MB-231 and MCF-7 cells were transfected with either si-CTL or si-ERRα for 48 h, followed by treatment with vehicle or cholesterol (5 µM for MDA-MB-231 cells, and 10 µM for MCF-7 cells) for 24 h. The above-mentioned kit was used according to the manufacturer’s protocol, and the NADPH concentration was determined colourimetrically based on the absorbance at 450 nm. All the data were normalized to the respective vehicle.

### 4.9. Cell Proliferation Assay

To determine whether the impact of cholesterol on TNBC-PDX cell proliferation is ERRα- dependent, the MTS Cell Proliferation Assay kit (ab197010, Abcam (Cambridge, MA, USA)) was utilized to assay the cell proliferation of TNBC-PDX cells. Based on the manufacturer’s instructions, TNBC-PDX cells were plated at a density of l0^4^ cells per well in 96-well plates. The cells were treated with a 10 μM concentration of cholesterol and/or cpd29. The medium was changed with the fresh medium containing the treatment every 48 h throughout the six days of the experiment. Per well, 20 μL of MTS reagent was then added and was incubated for one hour at 37 °C under standard culture conditions. The optical density (OD) value was determined at 490 nm using a microplate reader (Infinite M200PRO, TECAN (Morrisville, NC, USA)).

### 4.10. The Cancer Genome Atlas Analyses (TCGA)

The breast cancer gene expression data and their correlation to ERRα gene expression levels were obtained from TCGA datasets. Data were downloaded and visualized using the UALCAN web-portal at http://ualcan.path.uab.edu (May 2020) [78].

### 4.11. ERRα Signature Analysis in Basal-Like Overweight/Obese Breast Cancer Patients’ Primary Tumours vs. Lean Ones

To determine the ERRα gene expression signature profiles in basal-like overweight/obese breast cancer tumours in patients with BMI > 25 compared to those in patients with BMI ≤ 25 (lean), we analyzed the gene expression datasets from GEO: GSE78958 using the R interface. According to the GSE78958 study description, the gene expression data were generated using Affymetrix U133 2.0 gene expression for primary breast tumours, whereby their RNA was isolated from laser microdissected tissues [79]. The basal-like primary breast tumours were categorized based on the patients’ BMI (BMI > 25 considered as overweight/obese and BMI ≤ 25 considered as lean). We further stratified the data based on their ERRα expression levels, for analyzing the gene expression profiles. The patients’ ID used in this study is indicated in Appendix A for the basal-like subtype of breast cancer.

### 4.12. Statistical Analysis

All values are expressed as means of at least three independent experiments ± SEM. A two-tailed Student *t*-test was used to analyze the statistical significance of differences between two experimental groups, and two-way ANOVA was used to analyze comparisons between more than two groups. The experiments were repeated at least three times to obtain *p*-values. * represents *p* < 0.05 and was considered to be statistically significant. The data were plotted using GraphPad Prism (version 8.0., GraphPad Software Inc., San Diego, CA, USA) or R software (version 4.0.2, R core team, Vienna, Austria).

## 5. Conclusions

Our findings suggest that obesity modulates the metabolic gene expression profile of ERRα, a potent modulator of cellular metabolism, in breast cancer patients, in a manner consistent with the high cholesterol-induced alterations in ERRα-dependent metabolic activity in vitro in breast cancer cells. This, together with the reports that ERRα expression levels are high in primary breast tumours, particularly in TNBC, is associated with adverse clinical outcome [39,63,70,80].

Here, we provide a possible mechanistic explanation for the association between cholesterol/obesity and adverse metabolic reprogramming in breast cancer patients. In addition, targeting the metabolic vulnerabilities revealed in the current study under conditions of high cholesterol/obesity may lead to novel therapeutic strategies for this group of breast cancer patients. Furthermore, our findings support the notion of a novel combinational therapy that targets ERRα or ERRα-mediated metabolic vulnerability and cholesterol synthesis to treat breast cancer.

## Figures and Tables

**Figure 1 cancers-13-02605-f001:**
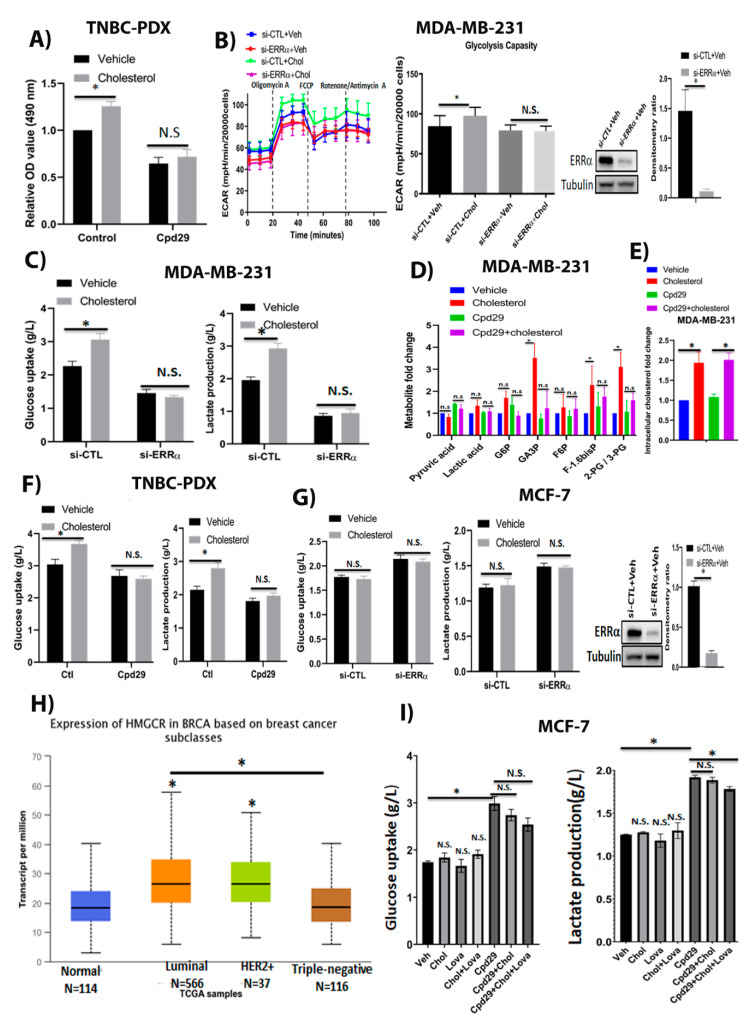
Cholesterol promotes cellular growth in TNBC-PDX cells, and enhances the abundances of glycolytic metabolites in TNBC cells, but not in ER+ breast cancer cells, in an ERRα-dependent manner. (**A**) TNBC-PDX cell proliferation was measured using an MTS kit in the presence of vehicle (Veh), cholesterol (Chol, 10 μM), and/or cpd29 (10 μM) on day 6. Ctl (control) represents the breast cancer cells treated with cholesterol or vehicle. (**B**) MDA-MB-231 cells were transfected with siRNA-control (si-CTL) or siRNA-ERRα (si-ERRα) and then treated with vehicle (Veh) or cholesterol (Chol, 5 μM) for 24 h. The extracellular acidification rate (ECAR) was obtained using the Seahorse XF96. Glycolysis capacity was measured after oligomycin drug injection. MDA-MB-231 cells were transfected with either siRNA-control (si-CTL) or siRNA-ERRα (si-ERRα) for 48 h, followed by treatment with vehicle or cholesterol 5 µM for 24 h. Cell lysates were immunoblotted using an anti-ERRα antibody. The densitometry ratio was measured using Image-J software (version 1.51, National Institutes of Health (NIH), Bethesda, MD, USA). (**C**) The glucose consumption and lactate production for MDA-MB-231 cells were obtained using a Profile 400 analyzer. The cells were transfected and treated as above. (**D**) The metabolic profiling on the glycolysis pathway was carried out in MDA-MB-231 cells using LC/MS. The data were normalized for cell count and the respective vehicle. The cells were treated with vehicle (Veh), cholesterol (Chol, 5 μM), and/or cpd29 (5 μM), which is an ERRα inhibitor, for 48 h. (**E**) Intracellular cholesterol levels were measured in MDA-MB-231 cells using GC/MS. The data were normalized for cell count and the respective vehicle. MDA-MB-231 cells were treated with vehicle (Veh), cholesterol (Chol, 5 μM), and/or cpd29 (5 μM) for 48 h. (**F**) Glucose uptake and lactate levels were measured in triple-negative breast cancer patient-derived xenograft (TNBC-PDX) cells. The cells were treated with vehicle (Veh), cholesterol (Chol, 10 μM), and/or cpd29 (10 μM) for 48 h; Ctl (control) represents the breast cancer cells treated with cholesterol or vehicle. (**G**) MCF-7 cells were transfected with siRNA-control (si-CTL) or siRNA-ERRα (si-ERRα), and then treated with vehicle or cholesterol (10 μM) for 24 h. MCF-7 cells were transfected with either siRNA-control (si-CTL) or siRNA-ERRα (si-ERRα) for 48 h, followede by treatment with vehicle or cholesterol 10 µM for 24 h. Cell lysates were immunoblotted using an anti-ERRα antibody. The densitometry ratio was measured using Image-J software. (**H**) Expression of HMGCR in various breast cancer subtypes, representing the mRNA levels of HMGCR gene in breast tumours and corresponding normal tissue obtained from the TCGA database. The data were visualized using the UALCAN web-portal. The significance (*) was defined by comparing each subtype to the normal tissue. Additionally, (*) shows significance between luminal and triple-negative tumour subtypes. (**I**) MCF-7 cells were treated with vehicle (Veh) or 10µM of cholesterol (Chol) and/or lovastatin (Lova) and/or cpd29 for 48 h. The results represent three independent experiments. A value of *p* < 0.05 was considered significant (*). N.S./n.s. = not significant.

**Figure 2 cancers-13-02605-f002:**
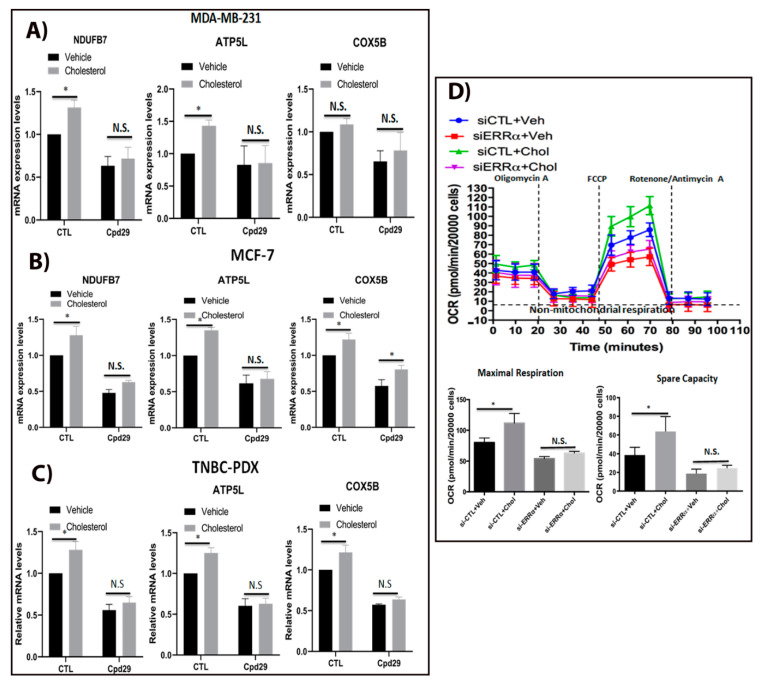
Cholesterol induces cellular respiration in ER+ and TNBC cells via ERRα. (**A**–**C**) MDA-MB-231 cells were treated with vehicle (Veh), cholesterol (Chol, 5 μM), and/or cpd29 (5 μM) for 48 h. Additionally, MCF-7 and TNBC-PDX cells were treated with vehicle (Veh), cholesterol (Chol, 10 μM), and/or cpd29 (10 μM) for 48 h. Total RNA was extracted and analyzed by RT-qPCR. Genes detected include: NDUFB7: NADH dehydrogenase (ubiquinone) 1 beta subcomplex subunit 7, ATP5L: ATP synthase subunit g (mitochondrial), and COX5B: Cytochrome *c* oxidase subunit 5B (mitochondrial). The mRNA data were normalized to endogenous GAPDH. Ctl (control) represents the breast cancer cells treated with cholesterol or vehicle. (**D**) Oxygen consumption rate (OCR) in MDA-MB-231 cells was obtained by Seahorse XF96. The cells were transfected with siRNA-control (si-CTL) or siRNA-ERRα (si-ERRα) and then treated with vehicle (Veh) or cholesterol (Chol, 5 μM) for 24 h. The maximal respiratory capacity (Maximal Resp.) represents the peak between FCCP and rotenone/antimycin A injection. Spare capacity was calculated by subtracting the maximal respiration from the basal respiration, as indicated in the graph. The data are expressed as means ± SEM, and represent at least three independent experiments. *p*-value < 0.05 was considered as significant (*). N.S./n.s. = not significant.

**Figure 3 cancers-13-02605-f003:**
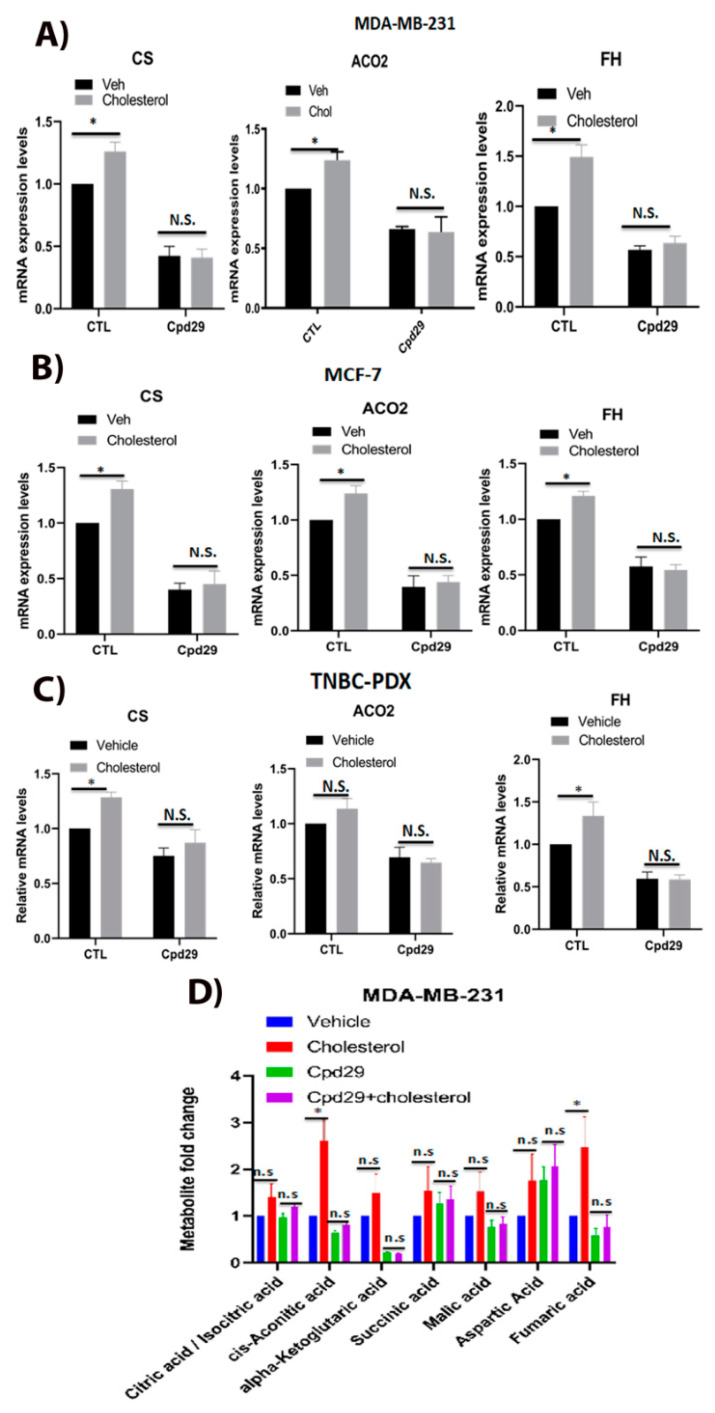
Cholesterol enhances the levels of three key enzymes and the abundance of two metabolites of TCA cycle in an ERRα-dependent manner. (**A**–**C**) MDA-MB-231 cells were treated with vehicle (Veh), cholesterol (Chol, 5 μM), and/or cpd29 (5 μM) for 48 h. Additionally, MCF-7 and TNBC-PDX cells were treated with vehicle (Veh), cholesterol (Chol, 10 μM), and/or cpd29 (10 μM) for 48 h. Total RNA was extracted and analyzed using RT-qPCR. The genes analyzed: CS: citrate synthase, mitochondria, ACO2: aconitase 2, mitochondria, and FH: fumarate hydratase. The mRNA data were normalized to endogenous GAPDH. Ctl (control) represents the breast cancer cells treated with cholesterol or vehicle in the absence of cpd29. (**D**) The metabolic profiling of TCA cycle intermediates was carried out using LC/MS. The data were normalized for cell count and the respective vehicle. MDA-MB-231 cells were treated with vehicle (Veh), cholesterol (Chol, 5 μM), and/or cpd29 (5 μM), which is an ERRα inhibitor, for 48 h. The data are expressed as mean ± SEM and represent at least three independent experiments. *p*-value < 0.05 was considered as significant (*). N.S./n.s. = not significant.

**Figure 4 cancers-13-02605-f004:**
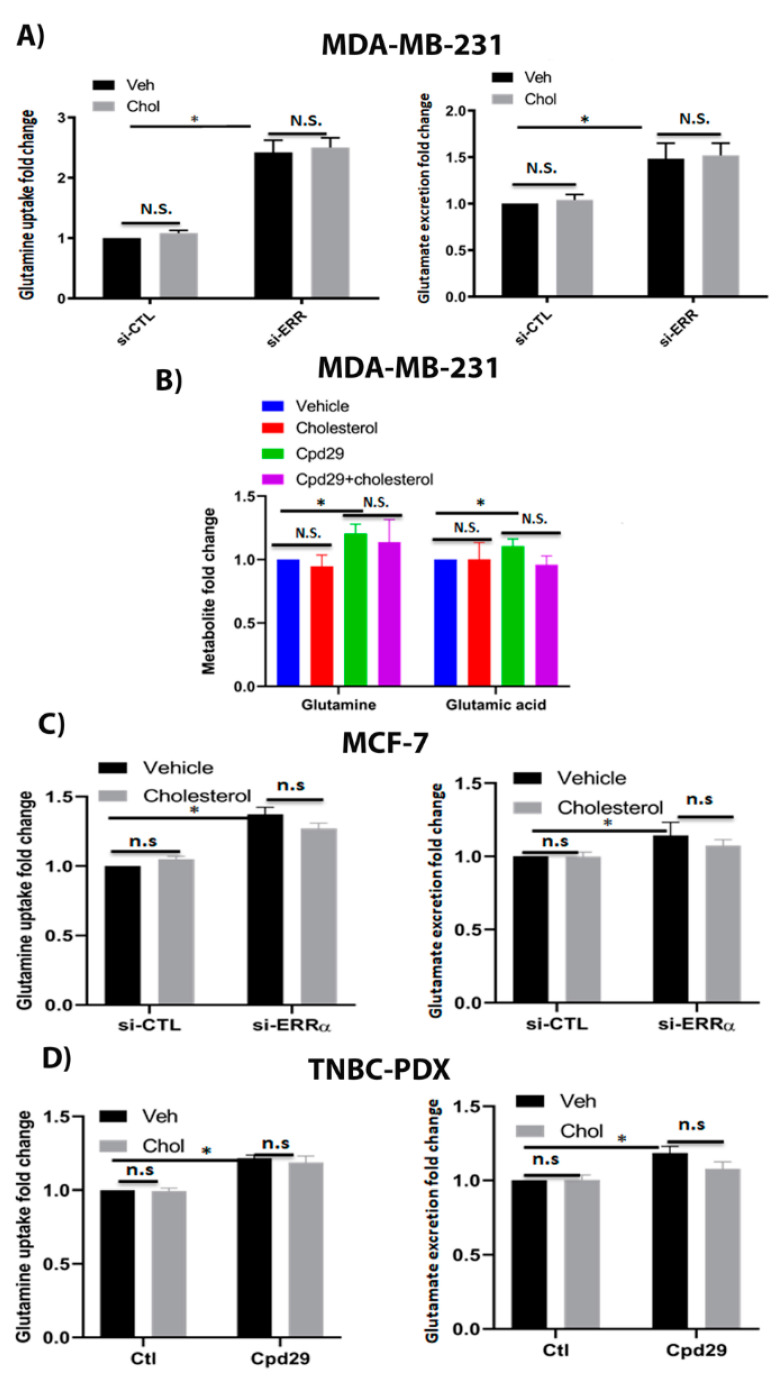
Cholesterol does not alter the glutaminolysis metabolite levels in breast cancer cells. (**A**) The glutamine uptake and glutamate and ammonium production were measured using a Profile 400 analyzer. MDA-MB-231 cells were transfected with siRNA-control (si-CTL) or siRNA-ERRα (si-ERRα) and then treated with vehicle (Veh) or cholesterol (Chol, 5 μM) for 24 h. (**B**) The metabolite levels were measured using GC/MS. MDA-MB-231 cells were treated with vehicle (Veh), cholesterol (Chol, 5 μM), and/or cpd29 (5 μM) for 48 h. (**C**) MCF-7 cells were transfected with siRNA-control (si-CTL) or siRNA-ERRα (si-ERRα) and then treated with vehicle (Veh) or cholesterol (Chol, 10 μM) for 24 h. (**D**) TNBC-PDX cells were treated with vehicle (Veh), cholesterol (Chol, 10 μM), and/or cpd29 (10 μM) for 48 h. Ctl (control) represents the breast cancer cells treated with cholesterol or vehicle. The results represent three independent experiments. A value of *p* < 0.05 was considered significant (*). N.S./n.s. = not significant.

**Figure 5 cancers-13-02605-f005:**
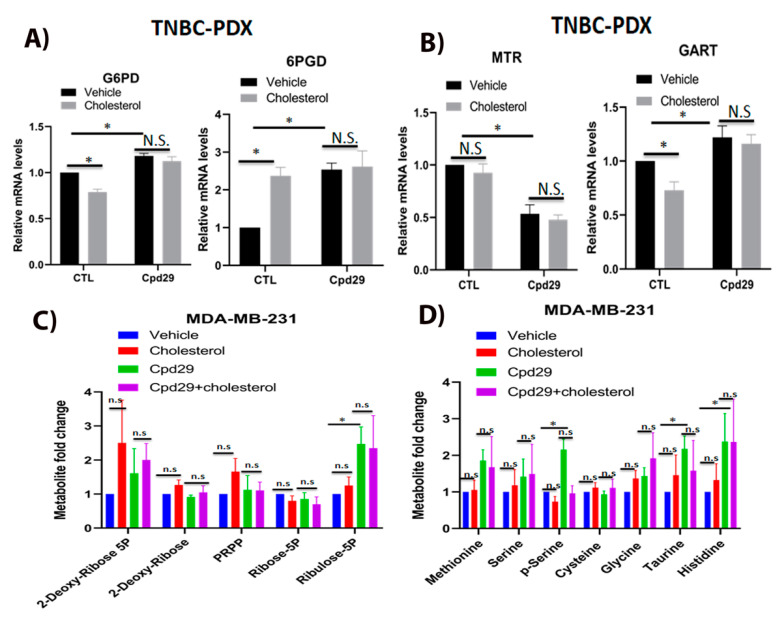
Cholesterol does not alter the metabolite levels involved in PPP and OCM pathways in TNBC cells. (**A**) TNBC-PDX cells were treated with vehicle (Veh), cholesterol (Chol, 10 μM), and/or cpd29 (10 μM) for 48 h. Total RNA was extracted and analyzed using RT-qPCR. The genes involved in PPP included: G6PD: glucose-6-phosphate 1-dehydrogenase, 6PGD: 6-phosphogluconate dehydrogenase. (**B**) TNBC-PDX cells were treated with vehicle (Veh), cholesterol (Chol, 10 μM), and/or cpd29 (10 μM) for 48 h. Total RNA was extracted and analyzed using RT-qPCR. The genes involved in the OCM pathway included: MTR: 5-methyltetrahydrofolate-homocysteine methyltransferase, GART: trifunctional purine biosynthetic protein adenosine-3. The mRNA data were normalized to endogenous GAPDH. CTL (control) represents the breast cancer cells treated with cholesterol or vehicle. (**C**) The metabolic profiling on PPP intermediates was carried out using LC/MS. The data were normalized for cell count and the respective vehicle. MDA-MB-231 cells were treated with vehicle (Veh), cholesterol (Chol, 5 μM), and/or cpd29 (5 μM), for 48 h. (**D**) The metabolic profiling on OCM intermediates was carried out using GC/MS. The data were normalized for cell count and the respective vehicle. MDA-MB-231 cells were treated with vehicle (Veh), cholesterol (Chol, 5 μM), and/or cpd29 (5 μM) for 48 h. The data are represented as means ± SEM, and at least three independent experiments. *p*-value < 0.05 was considered as significant (*). N.S./n.s. = not significant.

**Figure 6 cancers-13-02605-f006:**
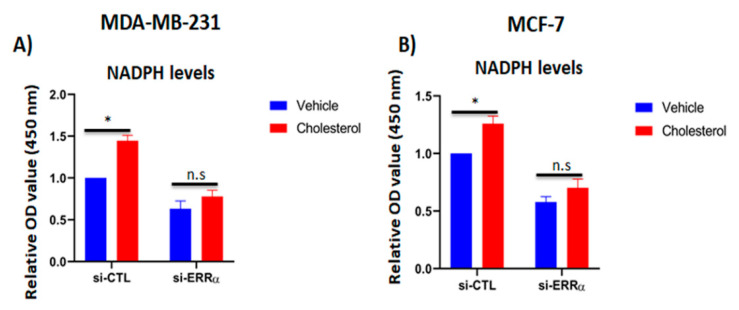
Cholesterol increases NADPH levels in breast cancer cells via ERRα pathway. (**A**,**B**) The intracellular NADPH levels were quantified using the NADP^+^/NADPH assay kit. MDA-MB-231 and MCF-7 cells were transfected with either siRNA-control (si-CTL) or siRNA-ERRα (si-ERRα) for 48 h, followed by treatment with vehicle or cholesterol (5 µM for MDA-MB-231 cells, and 10 µM for MCF-7 cells) for 24 h. A value of *p* < 0.05 was considered significant (*). N.S./n.s. = not significant.

**Figure 7 cancers-13-02605-f007:**
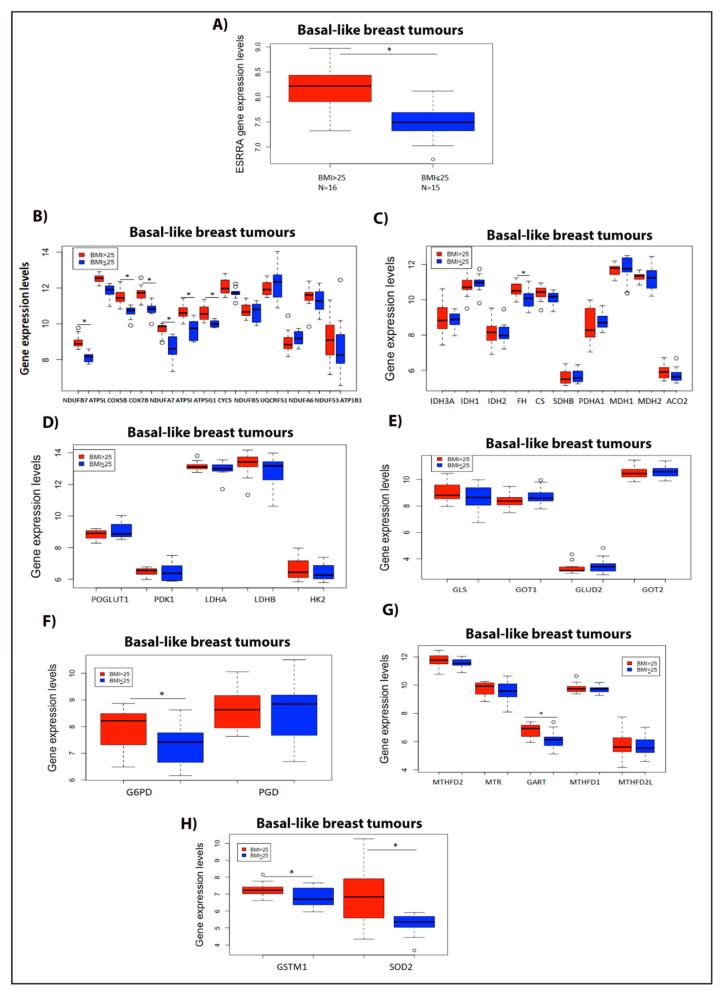
ERRα metabolic gene signature profile increases in basal-like breast tumours of overweight/obese vs. lean patients. (**A**) ESRRA gene expression levels in overweight/obese breast cancer patients with BMI > 25 (*N* = 16) vs. lean patients with BMI ≤ 25 (*N* = 15). (**B**) ERRα metabolic target genes involved in the OXPHOS pathway. (**C**) ERRα metabolic target genes associated with the TCA cycle. (**D**) ERRα metabolic target genes linked to the glycolytic pathway. (**E**) ERRα metabolic target genes related to the glutaminolysis pathway. (**F**) Gene expression levels involved in PPP intermediates. (**G**) ERRα metabolic target genes associated with the OCM pathway. (**H**) ERRα metabolic target genes involved in detoxifying enzymes. These data were obtained from the GEO: GSE78958 study, and were analyzed using R software (version 4.0.2, R core team, Vienna, Austria). For all the above panels, the number of patients in the BMI > 25 group is *N* = 16, and for the BMI ≤ 25 group is *N* = 15, and all the analyses were performed on basal-like breast tumours obtained from both BMI groups. *, significant (*p* < 0.05). The circle indicates the outliers in the datasets.

## Data Availability

The data are available from the corresponding author upon request.

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
