# Peer review of "Cholesterol-Induced Metabolic Reprogramming in Breast Cancer Cells Is Mediated via the ERRα Pathway"

_cancers, 2021, doi:10.3390/cancers13112605_

Round 1

Reviewer 1 Report

This manuscript investigated the effects of cholesterol in metabolic alteration in several breast cancer cells. By gene depletion and pharmacological inhibition, the authors demonstrated the metabolic reprogramming induction depends on the ERRa pathway.  In general, these data are interesting via metabolic profiling between different treatments in cells which can provide comprehensive insight into what pathways have been changed. There are several concerns from the reviewer as listed below.

  1. Based on the metabolic profiling data, does cholesterol treatment also increase the lipid metabolism and mitochondrial fatty acid oxidation in these breast cancer cells? The underneath mechanism of how cholesterol induces metabolic change and its connection with the ERRa pathway is undetermined. Does cholesterol treatment in these cells can induce ERRa expression?
  2. Figures are too small and in very low resolution. All figures should be enlarged to an appropriate scale and increase the resolution.
  3. Page 2, there is no data to support the result in the first sentence of paragraph 2.1.
  4. Page 3, add an introduction or description of cpd29 when the first time of describing it, without these, the reader is hard to know the background and what the purpose to use cpd29 here.
  5. According to Figure 1G, HMGCR mRNA expression is relatively higher in luminal and HER2+ patient tumors compared with normal or triple-negative subtypes. Based on the cell line models used in this study, MDA-MB-231 (TNBC) and MCF-7 (Luminal), the conclusion in Figure 1G should be made to compare with TNBC tumors.
  6. Figure 1A, The ECAR experiments usually start with treating cells with glucose, then followed with oligomycin and 2-DG. The compounds used in Figure 1A are more commonly used to measure the oxygen consumption rate or OCR in mitochondria. Please explain why there are some differences with the standard protocol of the seahorse experiment.
  7. It's hard to conclude from Figure 3A that cholesterol treatment could really increase the TCA cycle intermediates because most of the data was not reached statistical significance. How many samples measured for each group? Either tone down the conclusion or increase the sample number should address this problem.
  8. Page 8, 2.4, To compare the glutaminolysis rate, the glutamate to glutamine ratio should be calculated based on the intracellular metabolites levels among the different treatment groups.
  9. In Figure 5A and C, again, this data is not significantly suggesting the cholesterol treatment changes the PPP pathway or one-carbon metabolism, abundant level of some of the metabolites still showed an increasing trend, which is similar to the data in Figure 3A. However, the authors made different conclusions based on these results.
  10. Patient numbers and what public dataset was used for the analysis should be indicated in each panel of Figure 7.

Author Response

Reviewer 1

  1. Based on the metabolic profiling data, does cholesterol treatment also increase the lipid metabolism and mitochondrial fatty acid oxidation in these breast cancer cells? The underneath mechanism of how cholesterol induces metabolic change and its connection with the ERRa pathway is undetermined. Does cholesterol treatment in these cells can induce ERRa expression?

Response: Thank you for the interesting comment. The question as to whether cholesterol treatment also increases the lipid metabolism and mitochondrial fatty acid oxidation in these breast cancer cells is an interesting point and worthy of further investigation. Unfortunately, in the current study, the data obtained from LC/MS for the metabolites related to lipid metabolism and fatty acid oxidation were inadequate for analysis.  

Although the mechanism by which cholesterol induces metabolic changes and its connection with the ERRa pathway have not been fully elucidated, in addition to the data presented in the current study, we have previously reported some relevant findings (Ghanbari, Mader, & Philip: Cells, 9(8):1765, 2020). Our previous findings included demonstrating that (i)   cholesterol is an endogenous ligand of ERRa and that cholesterol directly and specifically binds to the ligand-binding-domain of ERRa with the relative Kd around 200 nM; (ii) exogenous cholesterol increases ERRa transcriptional activity and enhances the interaction of ERRa with its coactivator PGC-1a in breast cancer cells; (iii) cholesterol induces the expression of metabolic target genes of ERRa, such as IDH3A, PDK4, SOD2, GSTM1 and VEGF, in an ERRa-dependent manner. These results, together with the findings from the current study, suggest that cholesterol may induce metabolic changes by regulating the expression of the above metabolic target genes via the ERRa pathway.

We have previously reported that cholesterol treatment leads to an increase in the mRNA and protein levels of ERRa in breast cancer cells, likely via autoinduction (Fig 3 in Ghanbari et al: Cells, 9(8):1765, 2020). This is in agreement with the results in the current manuscript (Fig S1A). The above findings are mentioned and better clarified in the ‘Discussion’ section (pages 15&17). 

  1. Figures are too small and in very low resolution. All figures should be enlarged to an appropriate scale and increase the resolution.

Response: Thank you for the comment. We have enlarged the figures, and the resolution of the figures are 600 dpi.

  1. Page 2, there is no data to support the result in the first sentence of paragraph 2.1.

Response: The original first sentence of paragraph 2.1 that the reviewer referred to (currently the 2nd paragraph in Section 2.1) is now corrected to read as follows: “Next, we examined whether cholesterol regulates aerobic glycolysis in breast cancer cells in an ERRa-dependent manner. For this, we determined the levels of glycolytic metabolites in MDA-MB-231, and glucose uptake and lactate production levels in MDA-MB-231, TNBC-PDX, and MCF-7 cells, in which ERRa was knocked down using siRNA or ERRa activity was blocked with cpd29, followed by treatment with cholesterol.” (Section 2.1 has been rearranged with the first paragraph describing cellular proliferation, and a brief description on cpd29).  

The data demonstrating that cholesterol regulates aerobic glycolysis, glycolytic metabolites, glucose uptake, and lactate production in breast cancer (MDA-MB-231, MCF-7 or TNBC-PDX) cells in an ERRa-dependent manner are presented in figures 1B-1D, 1F, 1G&1I.

  1. Page 3, add an introduction or description of cpd29 when the first time of describing it, without these, the reader is hard to know the background and what the purpose to use cpd29 here.

Response: We agree that it is important to describe cpd29, the first time it appears in the text.  The description that  “Compound 29 (cpd29) is a selective inverse agonist of ERRa (Patch et al., 2011) and that it has been extensively used to inhibit ERRa transcriptional activity in in vitro and in vivo studies (Park et al., 2016; Park et al., 2019; Vernier et al., 2020) ”  have now been added in the ‘Results” section (section 2.1, page 2).

  1. According to Figure 1G, HMGCR mRNA expression is relatively higher in luminal and HER2+ patient tumors compared with normal or triple-negative subtypes. Based on the cell line models used in this study, MDA-MB-231 (TNBC) and MCF-7 (Luminal), the conclusion in Figure 1G should be made to compare with TNBC tumors.

Response: Thank you for the comment. We corrected the conclusion for Figure 1G (currently Fig 1H) to “Based on the TCGA database, the gene expression levels of 3-hydroxy-3-methyl-glutaryl-CoA reductase (HMGCR), a rate-limiting enzyme involved in the cholesterol biosynthesis pathway, are significantly higher in luminal breast tumors compared to the TNBC tumors (Figure 1G/currently Fig 1H)”. The text, legend, and the statistical analysis for Figure 1G/1H, were also corrected accordingly.

  1. Figure 1A, The ECAR experiments usually start with treating cells with glucose, then followed with oligomycin and 2-DG. The compounds used in Figure 1A are more commonly used to measure the oxygen consumption rate or OCR in mitochondria. Please explain why there are some differences with the standard protocol of the seahorse experiment.

Response: We agree that for ECAR experiment, a glycolytic stress kit, which contains glucose, oligomycin and 2-DG drugs, is what is commonly used. In the current manuscript, the glycolytic capacity was calculated after oligomycin drug injection. In addition, in Figure 1, we validated the ECAR result by measuring glucose uptake and lactate production which is associated with glycolysis pathway. In addition, we have measured the abundances of glycolytic metabolites. This approach has been used by others to determine both ECAR and OCR in previous publications  (Nicholas et al., 2017; Sonntag et al., 2017; Tan, Xiao, Li, Zeng, & Yin, 2015).

  1. It's hard to conclude from Figure 3A that cholesterol treatment could really increase the TCA cycle intermediates because most of the data was not reached statistical significance. How many samples measured for each group? Either tone down the conclusion or increase the sample number should address this problem.

Response: We agree with the reviewer that while cholesterol treatment leads to an increasing trend in the levels of most of the TCA cycle intermediates, this increase does not reach statistical significance, due to a large standard of error of the mean (4 samples per group were used). However, interestingly, our data demonstrate that exogenous cholesterol significantly increases the expression of three key TCA cycle enzymes FH, ACO2, and CS (as demonstrated in the current study) and IDH3A (as shown in a previously published study, Ghanbari et al, 2020), that catalyze the formation of the TCA cycle intermediates, in breast cancer cells, in an ERRa-dependent manner. Also, the exogenous cholesterol increases the abundance of two important TCA cycle intermediates. We now clarified the title in the Results section (2.3) to read: “Cholesterol enhances the abundance of certain key enzymes and metabolites of TCA cycle in an ERRa-dependent manner”.

  1. 8. Page 8, 2.4, To compare the glutaminolysis rate, the glutamate to glutamine ratio should be calculated based on the intracellular metabolites levels among the different treatment groups.

Response: Thank you for the comment. We agree that it is important to calculate the glutamate to glutamine ratio based on their intracellular levels.  However, for the purposes of the current paper, we were interested in determining whether cholesterol alters the metabolite levels involved in the glutaminolysis pathway. Based on the analysis data not shown here, converting the data to glutamate to glutamine ratio does not show any changes among the treatment groups. This is now clarified in the text (Result section, 2.4).

9.In Figure 5A and C, again, this data is not significantly suggesting the cholesterol treatment changes the PPP pathway or one-carbon metabolism, abundant level of some of the metabolites still showed an increasing trend, which is similar to the data in Figure 3A. However, the authors made different conclusions based on these results.

Response: We agree with the reviewer that although there is an increasing trend in the levels of certain metabolites involved in PPP and OCM pathway, those alterations were not statistically significant. The text in Section 2.5 now clearly states that “Cholesterol significantly decreases the expression of G6PD and GART in an ERRa-dependent manner and does not significantly alter the abundances of metabolites involved in the PPP and the OCM pathway, with the exception that cholesterol significantly increases 6PGD expression levels, possibly to maintain NADPH homeostasis.”

10.Patient numbers and what public dataset was used for the analysis should be indicated in each panel of Figure.

Response: Thank you for the important comment. We have now added the number of patients and the information on the public dataset to the Figure legend for each panel in Figure 7. The information (public dataset GSE78958 and the number of patients with the patient IDs) is also indicated in the Method section 4.11, and also in supplementary Table S2 for each group used for Figure 7.

References:

Ghanbari, F., Mader, S., & Philip, A. (2020). Cholesterol as an Endogenous Ligand of ERRα Promotes ERRα-Mediated Cellular Proliferation and Metabolic Target Gene Expression in Breast Cancer Cells. Cells, 9(8), 1765. doi:10.3390/cells9081765

Nicholas, D., Proctor, E. A., Raval, F. M., Ip, B. C., Habib, C., Ritou, E., . . . Nikolajczyk, B. S. (2017). Advances in the quantification of mitochondrial function in primary human immune cells through extracellular flux analysis. PLoS One, 12(2), e0170975. doi:10.1371/journal.pone.0170975

Park, S., Chang, C. Y., Safi, R., Liu, X., Baldi, R., Jasper, J. S., . . . McDonnell, D. P. (2016). ERRα-Regulated Lactate Metabolism Contributes to Resistance to Targeted Therapies in Breast Cancer. Cell Rep, 15(2), 323-335. doi:10.1016/j.celrep.2016.03.026

Park, S., Safi, R., Liu, X., Baldi, R., Liu, W., Liu, J., . . . McDonnell, D. P. (2019). Inhibition of ERRalpha Prevents Mitochondrial Pyruvate Uptake Exposing NADPH-Generating Pathways as Targetable Vulnerabilities in Breast Cancer. Cell Rep, 27(12), 3587-3601.e3584. doi:10.1016/j.celrep.2019.05.066

Patch, R. J., Searle, L. L., Kim, A. J., De, D., Zhu, X., Askari, H. B., . . . Gaul, M. D. (2011). Identification of Diaryl Ether-Based Ligands for Estrogen-Related Receptor α as Potential Antidiabetic Agents. J Med Chem, 54(3), 788-808. doi:10.1021/jm101063h

Sonntag, K.-C., Ryu, W.-I., Amirault, K. M., Healy, R. A., Siegel, A. J., McPhie, D. L., . . . Cohen, B. M. (2017). Late-onset Alzheimer’s disease is associated with inherent changes in bioenergetics profiles. Scientific Reports, 7(1), 14038. doi:10.1038/s41598-017-14420-x

Tan, B., Xiao, H., Li, F., Zeng, L., & Yin, Y. (2015). The profiles of mitochondrial respiration and glycolysis using extracellular flux analysis in porcine enterocyte IPEC-J2. Anim Nutr, 1(3), 239-243. doi:10.1016/j.aninu.2015.08.004

Vernier, M., Dufour, C. R., McGuirk, S., Scholtes, C., Li, X., Bourmeau, G., . . . Giguère, V. (2020). Estrogen-related receptors are targetable ROS sensors. Genes Dev, 34(7-8), 544-559. doi:10.1101/gad.330746.119

Wei, W., Schwaid, A. G., Wang, X., Wang, X., Chen, S., Chu, Q., . . . Wan, Y. (2016). Ligand Activation of ERRalpha by Cholesterol Mediates Statin and Bisphosphonate Effects. Cell Metab, 23(3), 479-491. doi:10.1016/j.cmet.2015.12.010

Reviewer 2 Report

The manuscript from Ghanbari and colleagues presented the outcomes of experiments using MDA-MB-231, MCF-7 and TNBC PDX cells, pre-treated with cholesterol supplementation of the culture media and EERa modulated pharmacologically or genetically. In this setting, the authors used a combination Seahorse, biochemistry and metabolomic analyses to assess cellular metabolism, assessed gene expression, as well as cell growth/viability with MTS assay. They also reported gene expression patterns in publicly available data when dichotomised by BMI – specifically <25 (i.e. lean) or >25 (overweigh-obese). In general, the authors demonstrate that ERRa function is required for many aspects of cholesterol-stimulated gene expression and cellular metabolism.

Overall, this is an interesting study; however, there are several significant issues with the current manuscript and interpretation of the data presented.

Major issues:

  1. I believe that the order of the data should be changed. The first data should be the demonstration that treating these cells with cholesterol enhanced cell proliferation and that this required ERRa (i.e. most of Fig 6). Then in each metabolic pathway, report the gene expression changes linked to this pattern before the metabolomic data. In other words, cholesterol stimulates cell growth due to ERRa regulated expression of genes involved in cellular metabolism and that these changes were functional in terms of metabolism. Further, evidence that the siRNA knockdown worked really must come in figure 1, not figure 6. I understand the rationale to split the metabolism data the way the authors have, but there is no linking between the sections, nor a strong rationale to report these pathways other than, we wanted to see if anything changed, when it should all be linked to cell proliferation.
  2. There are many studies that have shown that eating a high-cholesterol diet does not lead to increases in the levels of cholesterol in the circulation (PMID: 14074355 is one of the first and most elegant). As such, the authors need to describe this area of human physiology more accurately, and how it relates to breast cancer biology. Specifically, it is not appropriate to link studies that report cancer outcomes to cholesterol intake and those studies reporting a relationship between circulating cholesterol and cancer outcomes, as the link between cholesterol intake and circulating levels is not that simple.
  3. The authors should more accurately describe the experimental design as pretreatment with cholesterol. This is important as most measures were performed after 24 hours of cholesterol supplementation of culture media.
  4. Are the changes in the data presented expected when ERRa function was inhibited? There appear to be many meaningful changes in gene and/or metabolism variables between control and ERRa KD/ or cpd29 groups, without cholesterol treatment. This should be briefly discussed.
  5. What evidence do the authors have to show that cpd29 has inhibited ERRa activity, beyond the metabolic genes of interest? How do the authors know that this is a specific event and not just off-target that happens to modulate the genes of interest for the study? Similarly, what evidence do the authors have to show that Lovastatin reduced cholesterol levels in treated cells? Also, did the conditions used in Figure 1H correlates with cell growth?
  6. Are the changes in oxidative metabolism linked to mitochondrial biogenesis? Also, the authors must be cautious when interpreting the changes in steady state levels of metabolic intermediates and inferring changes in flux/enrichment.
  7. The authors state that they have used One-Way ANOVAs, yet in nearly all the graphs there are two variables – ERRa modulation and cholesterol treatment. As such, these data must be tested using a Two-Way ANOVA to determine the effect of the two variables.
  8. The data in figure 7 does not report differences between lean and obese patients, but lean vs overweight/obese. Further, is it possible to report other characteristics of this population, such as their circulating cholesterol or anything metabolic to link these changes to those that relate to the overarching themes of the study?
  9. The introduction set up the study quite nicely, but I’m not convinced that there is a need to cite eleven reviews that are very similar to the first dozen lines.
  10. The discussion is very long and excessive re-describing of the results should be avoided.
  11. How do the authors reconcile that they report that cholesterol did not alter PPP metabolite levels, yet NADPH levels were increased, and then claim in the discussion that the change in NADPH is likely linked to increased biomass via PPP and ROS detoxification (without measuring ROS)?

Minor issues:

  1. The figures are very small and not well put together. The quality of the image is very poor but that is a function of the MDPI submission process heavily reducing the size and quality of the image. The bordering of the sub-panels should be removed and the arrangement of each figure needs to be improved.
  2. The results section should be written in past tense.
  3. Please change the scale in the y-axis of Fig 1A so that it is less busy – i.e. make the major numbers every 20 or 25 rather than 10.
  4. The y-axis legend for glucose uptake needs to be changed so it is more accurate. It is a reduction in glucose levels in the media, normalized to Control. Likewise, lactate production has been normalized. I am not sure why these cannot be reported as absolute values, e.g. mM for glucose.
  5. Figure 4A is not normalised to 1.

Round 2

Reviewer 1 Report

The current version has been largely improved, the authors have addressed most of the concerns from the reviewer. 

Reviewer 2 Report

The authors have sufficiently addressed my concerns.